# Probing a leptophobic top-colour model
# with cross section measurements and precise
# signal and background predictions: A case study

**Mohammad Mahdi Altakach[1,2,3], Jonathan Mark Butterworth[4],**
**Tomáš Ježo[2], Michael Klasen[2] and Ingo Schienbein[1]**

**1** Laboratoire de Physique Subatomique et de Cosmologie, Université Grenoble-Alpes,
CNRS/IN2P3, 53 Avenue des Martyrs, 38026 Grenoble, France
**2** Institut für Theoretische Physik, Westfälische Wilhelms-Universität Münster,
Wilhelm-Klemm-Str. 9, 48149 Münster, Germany
**3** Institute of Theoretical Physics, Faculty of Physics, University of Warsaw,
ul. Pasteura 5, PL-02-093 Warsaw, Poland
**4** Department of Physics & Astronomy, UCL, Gower St., WC1E 6BT, London, UK

## Abstract

**The sensitivity of particle-level fiducial cross section measurements from ATLAS, CMS and LHCb to a leptophobic top-colour model is studied. The model has previously been the subject of resonance searches. Here we compare it directly to state-of-the-art predictions for Standard Model top quark production and also take into account next-to-leading order predictions for the new physics signal. We make use of the Contur framework to evaluate the sensitivity of the current measurements, first under the default Contur assumption that the measurement and the SM exactly coincide, and then using the full SM theory calculation for $t\bar{t}$ at next-to-leading and next-to-next-to-leading order as the background model. We derive exclusion limits, discuss the differences between these approaches, and compare to the limits from resonance searches by ATLAS and CMS.**

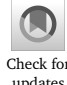

# 1   Introduction

The quest for physics that goes beyond the Standard Model (SM) of particle physics is one of the most important research goals of the Large Hadron Collider (LHC) at CERN, particularly after the great success of the Higgs boson discovery in 2012. Indeed, for at least the next 15 years the LHC will remain our best hope for discovering new physics in a controlled collider environment. During Run 2, the LHC has already collected data with an integrated luminosity of about 140 fb$^{-1}$ per experiment. During Run 3 (2022-2025) the statistics will be roughly doubled to 250 fb$^{-1}$, and during the High Luminosity LHC phase (HL-LHC) starting in 2029 it is expected that an integrated luminosity of up to 3000 fb$^{-1}$ will be reached. This will allow access to lower cross-sections, in particular to those high energy regions where differential cross sections decrease rapidly. The full exploitation of the future LHC data therefore remains one of the most important tasks in particle physics in the coming years.

Apart from a few promising hints, e.g. in rare decays of heavy $B$-mesons [1–4], however, no clear signs of new physics have so far appeared in any of the experimental analyses. Therefore, it becomes increasingly probable that any potential new physics effect at the LHC will be subtle, e.g. it may appear as a small deviation in kinematic distributions due to the influence of loop effects. As a consequence, precise theoretical predictions for observables in the SM and theories Beyond the SM (BSM) are very important. In view of the many null-results in the channel-by-channel searches, it becomes also mandatory to change perspective. Firstly, a more global approach is required, as opposed to benchmark-driven signature-by-signature searches. Secondly, the use of differential cross section measurements allows direct comparison to precision SM predictions. As well as facilitating such a global approach to discovering where BSM physics may hide, this will also allow the level of precision at which the SM describes those measurements to be quantified.

It is natural to perform global analyses in the context of an effective field theory (EFT) such as the SM EFT [5]. The advantage of this approach is that it is rather model-independent, so that a large variety of postulated BSM theories and scenarios can be efficiently constrained. On the other hand, in order for the EFT to be valid at LHC energies, the scale of new physics $\Lambda$ has to lie above the LHC energy scale, i.e. beyond the direct reach of the LHC. For this reason, a complementary direct approach remains relevant. Here, specific models are probed in the context of a global analysis of a variety of LHC data. One may then constrain the allowed parameter space of the model, or, in the case of clear deviations from the SM, analyse the likelihood of this specific BSM theory, without the restrictions on the applicability and the ambiguity of an EFT. Obviously the constraints themselves are model dependent; however, by making use of particle-level cross sections, the model-independence of the data is retained and so many models may be rapidly investigated with the same measurements.

In this study we follow the latter approach, using the Constraints On New Theories Using Rivet (Contur) toolkit [6, 7] to examine the sensitivity of ATLAS, CMS and LHCb particle-level fiducial cross section measurements, available in Rivet 3.1.4 [8], to a leptophobic top-colour [9, 10] scenario. Contur uses the measurements preserved in Rivet, a system for the validation and tuning of Monte Carlo event generators, in order to test new BSM models. There are a number of improvements with respect to previous analyses with Contur:

- This is the first Contur analysis using higher-order theory predictions for the SM background. Previous studies have used data as the background expectation. Since the measurements concerned have all been shown to agree with SM expectations, this is equivalent to assuming the SM uncertainties are negligible compared to the measurement uncertainties. The inclusion of the SM theory predictions for the relevant fiducial cross sections in the Contur framework, carried out as part of this work, allows us to examine the validity of this assumption.

- We also obtain next-to-leading order (NLO) predictions for the new physics signals. The relevant NLO calculations are consistently matched to parton shower Monte Carlo generators in the POWHEG box framework and include also electroweak contributions [11, 12]. Most Contur results to date have used the inclusive LO calculations of Herwig [13] for their signal predictions.

The top-colour model considered here (see Sec. 2) has been previously analysed in several experimental searches of new heavy spin-one resonances [14–20]. The fact that the signature is simply a resonance in the $t\bar{t}$ channel implies that the benefits of a global analysis are less clear than might be the case for models with a more complex phenomenology, or models which are less well studied. However, our purpose is to examine the direct use of precision SM calculations in probing BSM physics, and in this sense the model is a good test case, since higher order predictions for both signal and background are available. As such, this paper is a proof of concept exploring the possibility to extend the Contur idea to higher perturbative orders. For example, the calculation in Ref. [12] covers a wider class of models with $Z'$ and $W'$ resonances, which can be scanned in the future. Furthermore, the theory predictions for the SM background remain relevant also for other classes of models.

The paper is structured as follows: first, we discuss the calculations used, comparing the full NLO POWHEG calculation of $t\bar{t}$ production [12, 21–23] – the main process of interest – with the more inclusive, but LO, Herwig calculations based on the same model. We then evaluate the sensitivity of the current measurements, both under the default Contur assumptions that the measurement and the SM exactly coincide, and using the full SM theory calculation for $t\bar{t}$ as the background model and discuss the differences. We conclude with an estimate of the current exclusion limits and the potential future reach of LHC data.

## 2 Calculations of signal and background

In this section, we describe the theoretical framework of our POWHEG calculations for both the signal and background processes.

First, we employ the NLO LUXqed parton distribution functions (PDFs) obtained within the NNPDF3.1 global fit [24–26] as implemented in the LHAPDF library (ID = 324900) [27, 28]. This set provides, in addition to the quark and gluon PDFs, a precise determination of the photon PDF inside the proton, which we need for our predictions of electroweak cross section contributions. The PDF uncertainties are calculated using Eqs. (21) and (22) of Ref. [29].

Second, the strong coupling constant $\alpha_s(\mu_R)$ is evaluated at NLO in the $\overline{\text{MS}}$ scheme. It is provided together with the PDF set and satisfies the condition $\alpha_s(M_Z) = 0.118$. While our choices of renormalisation and factorisation scales depend on the considered subprocess, we always identify the two scales for our central predictions and evaluate the scale uncertainties with the usual seven-point method, i.e. by independently multiplying the scales by factors of $\xi_R, \xi_F \in \{0.5, 1, 2\}$ discarding combinations with $\xi_F/\xi_R = 4$ or $1/4$. For the total theoretical uncertainty on the SM cross section, we take the envelope of all predictions resulting from scale and PDF variations. This uncertainty is applied to the SM background calculations when evaluating the sensitivities with Contur, which treats the PDF and scale as correlated uncertainty sources within a given measurement, and sums them in quadrature with the statistical uncertainty. They are not applied to the signal calculations, where statistical uncertainties dominate. The setup described above is used throughout the rest of the publication unless specified otherwise.

## 2.1 Top-colour model signal

The fact that the top-quark mass is large indicates that it may play a special role with respect to electroweak symmetry breaking. One possibility to generate a large top-quark mass is provided by the so-called Top-Colour (TC) model [9, 10], where a top-quark pair condensate is dynamically generated by an additional strong SU(3) gauge group that couples only to the third generation, while the original SU(3) gauge group couples only to the first and second generations. The two groups can then be broken to the QCD group $SU(3)_C$ in order to restore the strong dynamics of the SM.

To prevent the formation of a bottom-quark condensate, an additional U(1) symmetry and associated $Z'$-boson must be introduced. In Ref. [30], four variants of the TC model are proposed, which correspond to four different choices of the couplings between the additional $Z'$-boson and the three fermion generations. We focus in this article on the Model IV of the reference cited above, which is known as the leptophobic TC model [31]. The $Z'$-boson in this model does not couple to the second generation of quarks and, as indicated by the name of the model, has no significant couplings to leptons.

The Lagrangian of the leptophobic TC model is given in Ref. [31] and reads

$$
\mathcal{L} = \left( \frac{1}{2} g_1 \cot \theta_H \right) Z'^\mu \big( \bar{t}_L \gamma_\mu t_L + \bar{b}_L \gamma_\mu b_L + f_1 \bar{t}_R \gamma_\mu t_R + f_2 \bar{b}_R \gamma_\mu b_R
$$
$$
- \bar{u}_L \gamma_\mu u_L - \bar{d}_L \gamma_\mu d_L - f_1 \bar{u}_R \gamma_\mu u_R - f_2 \bar{d}_R \gamma_\mu d_R \big).
\tag{1}
$$

Here, $g_1$ is the $U(1)_Y$ coupling constant of the SM hypercharge, $\cot \theta_H$ is the ratio of the two U(1) coupling constants, and $f_1$ and $f_2$ are the relative strengths of the couplings of right-handed up- and down-type quarks with respect to those of the left-handed quarks. We set $f_1$ and $f_2$ to 1 and 0, respectively. The parameter $\cot \theta_H$ is related to the total decay width of the $Z'$-boson, which is given in Ref. [31] as

$$
\Gamma_{Z'} = \frac{\alpha \cot^2 \theta_H M_{Z'}}{8 \cos^2 \theta_W} \left[ \sqrt{1 - \frac{4 M_t^2}{M_{Z'}^2}} \left( 2 + \frac{4 M_t^2}{M_{Z'}^2} \right) + 4 \right].
\tag{2}
$$

The TC signal is then calculated using our `PBZpWp` event generator [12, 21], where both the BSM production of top-quark pairs and the interference with the electroweak SM processes are implemented. Note that `PBZpWp` also provides predictions for the interference of BSM production with the SM QCD processes, but these contributions vanish in the TC model. The `PBZpWp` generator employs the POWHEG [32, 33] method within the POWHEG BOX framework [11, 34] and matches NLO calculations with parton showers (PS). For the TC signal, we set the factorisation and renormalisation scales to the partonic centre-of-mass energy, $\mu_F = \mu_R = \sqrt{\hat{s}}$. The top-quark decay, PS and modelling of non-perturbative effects are all performed by `Pythia 8.2` [35]. The mass of the $Z'$-boson is treated as a free parameter, as is $\cot \theta_H$, which in turn determines the width (see above).

For comparison with the `PBZpWp` results, we also use the `Herwig` event generator [13]. This method is less precise, being based on leading-order (LO) estimates, but is fast, and is the default method of evaluating potential signals in `Contur`. We have generated, using the UFO [36] model file for the TC model, all $2 \to 2$ diagrams involving a BSM particle either in the $s$-channel propagator or as an outgoing leg. In this case, there is no matching or merging between the PS and higher-order QCD diagrams. Instead, `Herwig` separates $s$-channel diagrams of the type $q\bar{q} \to Z' \to t\bar{t}$ from the QCD radiative diagrams $q\bar{q} \to Z'g$ (with subsequent decay of $Z' \to t\bar{t}$) using a transverse momentum cut, $k_\perp^{\min}$, on the radiated gluon. This approximate procedure can emulate the most important real emission part of the higher-order corrections to $s$-channel $Z'$-exchange, but will create double-counting with the PS and thus

overestimate the cross section, if $k_\perp^{\min}$ is too low. We therefore varied $k_\perp^{\min}$ from 10 GeV to 1 TeV, the default value being 20 GeV, and $M_{Z'}$ between 2 and 5 TeV. We find that the cross section for the $q\bar{q} \to Z'g$ subprocess drops below the $s$-channel process, for $k_\perp^{\min} \approx 100$ GeV. Furthermore, above about 50 GeV the Herwig calculation is in good agreement with POWHEG for the considered subprocess. We therefore use $k_\perp^{\min} = 50$ GeV in our Herwig studies. We use the CT14 [37] PDF set, which is the default in Herwig.

## 2.2 Standard Model background

In the SM, pairs of top quarks can be produced both strongly and electroweakly. The production modes due to electroweak forces are often neglected, as they are relatively suppressed by the small value of the corresponding coupling constant. However, in the BSM model considered here the new physics couples via electroweak-like couplings, so that we also have to consider SM electroweak $t\bar{t}$ production and its QCD corrections. To be more precise, we consider the QCD top-pair production to $\mathcal{O}(\alpha_S^2)$ and $\mathcal{O}(\alpha_S^3)$, electroweak top-pair production to $\mathcal{O}(\alpha^2)$ and $\mathcal{O}(\alpha^2 \alpha_S)$, and mixed production to $\mathcal{O}(\alpha \alpha_S)$. Conversely, we neither consider electroweak corrections to strong processes of $\mathcal{O}(\alpha_S^2)$, nor QCD corrections to mixed $\mathcal{O}(\alpha \alpha_S)$ processes, which are of the same order, nor non-resonant production modes that can yield the same final state as the resonant ones after both top quarks have decayed.

We simulate the QCD production of top-quark pairs up to NLO QCD using the hvq [23] event generator, which again matches NLO corrections to the PS using the POWHEG method. For the $s$-channel and $t$-channel electroweak production mediated by the $Z$- and $W$- bosons up to NLO QCD, we use our PBZpWp event generator. It also includes the mixed QCD and electroweak production, i.e. both the interference between the purely QCD and the purely electroweak production modes and the photon induced channels. For the SM background processes, the factorisation and renormalisation scales in both hvq and PBZpWp are identified with the transverse mass of the top quark in the rest frame of the $q\bar{q}$ system: $\mu_F = \mu_R = \sqrt{p_T^2 + M^2}$.

Higher order QCD corrections for top-pair production up to next-to-next-to-leading order (NNLO) have now been available for some time [38–41]. Recently, a method for matching such NNLO calculations to PS has been introduced in Ref. [42]. Additionally to the hvq event sample we consider the event sample of Ref. [42] which was obtained for the LHC operating at 13 TeV with the NNPDF31_nnlo_as_0118 (303600) PDF set. The renormalisation and factorisation scales in this sample are set to $\mu_F = \mu_R = 0.5 M_{t\bar{t}}$.

The decay of the top quark (if not already included in the event generator), the PS and the modelling of non-perturbative effects are, as in the signal case, carried out by Pythia 8.2.

Using the event generators mentioned above, thirteen different LHC measurements of top-quark pair production at both 8 TeV (NLO predictions only) and 13 TeV (NNLO and NLO) centre-of-mass energy were simulated [43–55]. In addition, we simulate the ATLAS inclusive jet and dijet cross section measurement [56] using the dijet [57] POWHEG package, where we use the default choice for the renormalisation and factorisation scales, i.e. the transverse momentum of the two jets in the underlying Born configuration. We set the minimum generation cut and the Born suppression parameter to 50 GeV and 1000 GeV, respectively. Again, the showering, the hadronisation and the multiparton interactions are performed using Pythia 8.2.

For convenience, all the employed data sets that we have higher-order theory predictions for are summarised in Tab. 1.

Table 1: Table of the Rivet routines used for the limit-setting scan. Only routines where we have higher-order theory predictions are listed. The number in the Contur category indicates the centre-of-mass energy. The calculations are performed by the authors unless otherwise cited.

| Contur Category | $\mathcal{L}$ [fb$^{-1}$] | Rivet/Inspire ID | Highest SM Order | Brief description |
|---|---|---|---|---|
| ATLAS 8 LMETJET<br>ATLAS $\ell+E_T^{\mathrm{miss}}$+jet | 20.3 | ATLAS_2015_I1397637 [44] | NLO | Boosted $t\bar{t}$ differential cross-section |
| ATLAS 8 LMETJET<br>ATLAS $\ell+E_T^{\mathrm{miss}}$+jet | 20.3 | ATLAS_2015_I1404878 [45] | NLO | $t\bar{t}$ (to l+jets) |
| CMS 8 LMETJET<br>CMS $\ell+E_T^{\mathrm{miss}}$+jet | 19.7 | CMS_2017_I1518399 [46] | NLO | $t\bar{t}$ as a function of the leading jet mass for boosted top |
| ATLAS 13 LMETJET<br>ATLAS $\ell+E_T^{\mathrm{miss}}$+jet | 3.2 | ATLAS_2017_I1614149 [50] | NNLO | Resolved and boosted $t\bar{t}$ l+jets |
| ATLAS 13 LMETJET<br>ATLAS $\ell+E_T^{\mathrm{miss}}$+jet | 3.2 | ATLAS_2018_I1656578 [51] | NNLO | Semileptonic $t\bar{t}$ |
| ATLAS 13 LMETJET<br>ATLAS $\ell+E_T^{\mathrm{miss}}$+jet | 36 | ATLAS_2019_I1750330 [49] | NNLO | Semileptonic $t\bar{t}$ |
| CMS 13 LMETJET<br>CMS $\ell+E_T^{\mathrm{miss}}$+jet | 2.3 | CMS_2016_I1491950 [54] | NNLO | Semileptonic $t\bar{t}$ |
| CMS 13 LMETJET<br>CMS $\ell+E_T^{\mathrm{miss}}$+jet | 35.9 | CMS_2018_I1662081 [52] | NNLO | Semileptonic $t\bar{t}$ |
| CMS 13 LMETJET<br>CMS $\ell+E_T^{\mathrm{miss}}$+jet | 35.8 | CMS_2018_I1663958 [43] | NNLO | $t\bar{t}$ lepton+jets |
| ATLAS 13 L1L2METJET<br>ATLAS $\ell_1\ell_2+E_T^{\mathrm{miss}}$+jet | 36.1 | ATLAS_2019_I1759875 [48] | NNLO | Dileptonic $t\bar{t}$ |
| LHCB 13 L1L2B[1]<br>LHCb $\ell_1\ell_2$+bb | 1.93 | LHCB_2018_I1662483 [55] | NLO | Forward top pair production in the dilepton channel |
| ATLAS 13 TTHAD<br>ATLAS Hadronic $t\bar{t}$ | 36.1 | ATLAS_2018_I1646686 [53] | NNLO | All-hadronic boosted $t\bar{t}$ |
| CMS 13 TTHAD<br>CMS Hadronic $t\bar{t}$ | 35.9 | CMS_2019_I1764472 [47] | NNLO | $t\bar{t}$ cross section as a function of the jet mass in boosted hadronic top quark decays |
| ATLAS 13 JETS<br>ATLAS jets | 3.2 | ATLAS_2018_I1634970 [56] | NLO | ATLAS inclusive jet and dijet cross sections |
| ATLAS 13 METJET<br>ATLAS $E_T^{\mathrm{miss}}$+ jets | 3.2 | ATLAS_2017_I1609448 [58] | NLO [59] | ATLAS $E_T^{\mathrm{miss}}$ measurement |
| ATLAS 8 EEJET<br>ATLAS $ee$+jet | 20.3 | ATLAS_2015_I1408516 [60] | NLO [61] | ATLAS de-electron pairs |
| ATLAS 8 MMJET<br>ATLAS $\mu\mu$+jet | 20.3 | ATLAS_2015_I1408516 [60] | NLO [61] | ATLAS dimuon pairs |

---

[1]Even though the LHCb measurement was used for the limit-setting scan, it was never the most sensitive one.

# 3 Sensitivity

## 3.1 Default background model

As discussed previously, the default `Contur` approach is to take the fact that all the measurements considered have been shown in their original publications to be consistent with the SM, and make the additional assumption that they are identical to it; the sensitivity is then derived by seeing how much room the experimental uncertainties leave for a BSM contribution, using a $\chi^2$ test to evaluate the relative likelihood, as discussed in Ref. [7]. The results using this approach, employing either POWHEG or Herwig for the signal, are shown in Fig. 1.

The general features are similar with both POWHEG and Herwig, with measurements involving tops giving the greatest sensitivity. At lower masses, several measurements have similar sensitivity, with the most sensitive at each point subject to statistical fluctuations, leading to a patterning in those regions of the figures. At high $M_{Z'}$, the boosted, fully hadronic top cross section gives the greatest sensitivity of the top measurements. However, especially in the Herwig case, the sensitivity extends to higher masses than for POWHEG, and this is driven by the ATLAS jet measurements at 13 TeV [56, 62]. This final state receives contributions not only from $q\bar{q} \to Z' \to t\bar{t}$, but also from $q\bar{q} \to Z' \to q\bar{q}$, where $q = u, d$. In the inclusive, but LO, generation of Herwig these are included, whereas in the NLO calculation of POWHEG only the $Z'$ decay to tops is implemented. Over the $\cot\theta_H$ range covered, the ratio of the width of the $Z'$ to its mass $M_{Z'}$ lies between 0.015 and 0.15. We will return to this aspect in Section 3.3, after first discussing the higher order top calculations in more detail.

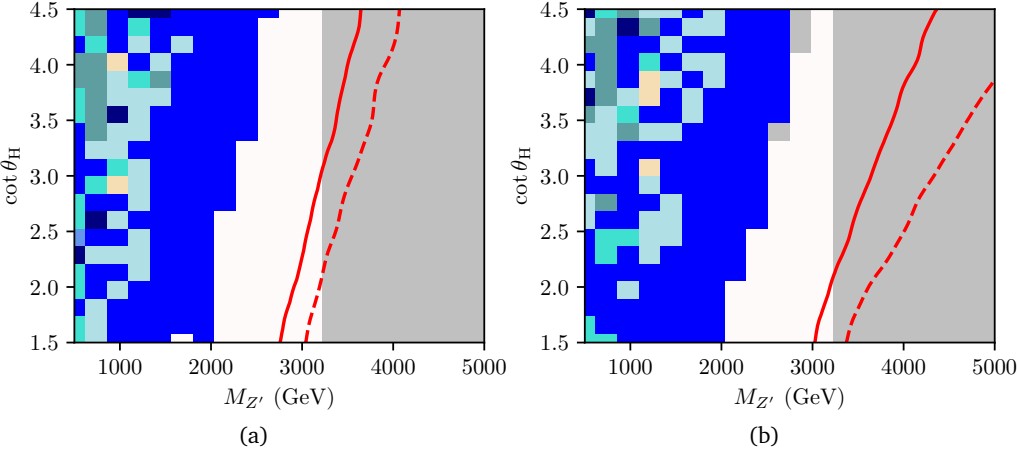

Figure 1: Sensitivity to the leptophobic TC model, in the $Z'$ mass (GeV) versus the $\cot\theta_H$ plane. The coloured blocks indicate the most sensitive final state (see legend below). The 95% CL (solid red) and 68% CL exclusion (dashed red) contours are superimposed, considering the data as background. (a) NLO $t\bar{t}$ signal calculated using POWHEG, (b) signal calculated using Herwig (inclusive LO).

| | | |
|---|---|---|
| CMS $\ell + E_T^{\text{miss}}$+jet | ATLAS $\ell + E_T^{\text{miss}}$+jet | ATLAS $e + E_T^{\text{miss}}$+jet |
| ATLAS $\mu + E_T^{\text{miss}}$+jet | ATLAS jets | CMS Hadronic $t\bar{t}$ |
| ATLAS Hadronic $t\bar{t}$ | ATLAS $\ell_1\ell_2 + E_T^{\text{miss}}$ | ATLAS $\ell_1\ell_2 + E_T^{\text{miss}}$+jet |

## 3.2 SM calculation as background

The higher order SM predictions for top final states, discussed in Section 2.2, can also be used directly as the background expectation by Contur when calculating the sensitivity.

Fig. 2 shows the sensitivity again, now in the plane of the ratio of the width of the $Z'$ to its mass $M_{Z'}$, versus $M_{Z'}$. For a given $M_{Z'}$, there is a one-to-one correspondence between $\cot\theta_{\mathrm{H}}$ and the $\Gamma_{Z'}$, given by eq. (2), with $\cot\theta_{\mathrm{H}} = 0.45$ corresponding to $\Gamma_{Z'}/M_{Z'} = 0.00154$, for $M_{Z'} = 2$ TeV. In Fig. 2a we again use the data as the SM background, but only use that subset of measurements for which Contur has access to the NLO SM predictions (See Table 1). This then allows a fair comparison with Fig. 2b, in which the NLO SM calculations are used as the background. It can be seen that the limits are similar, which is expected since the

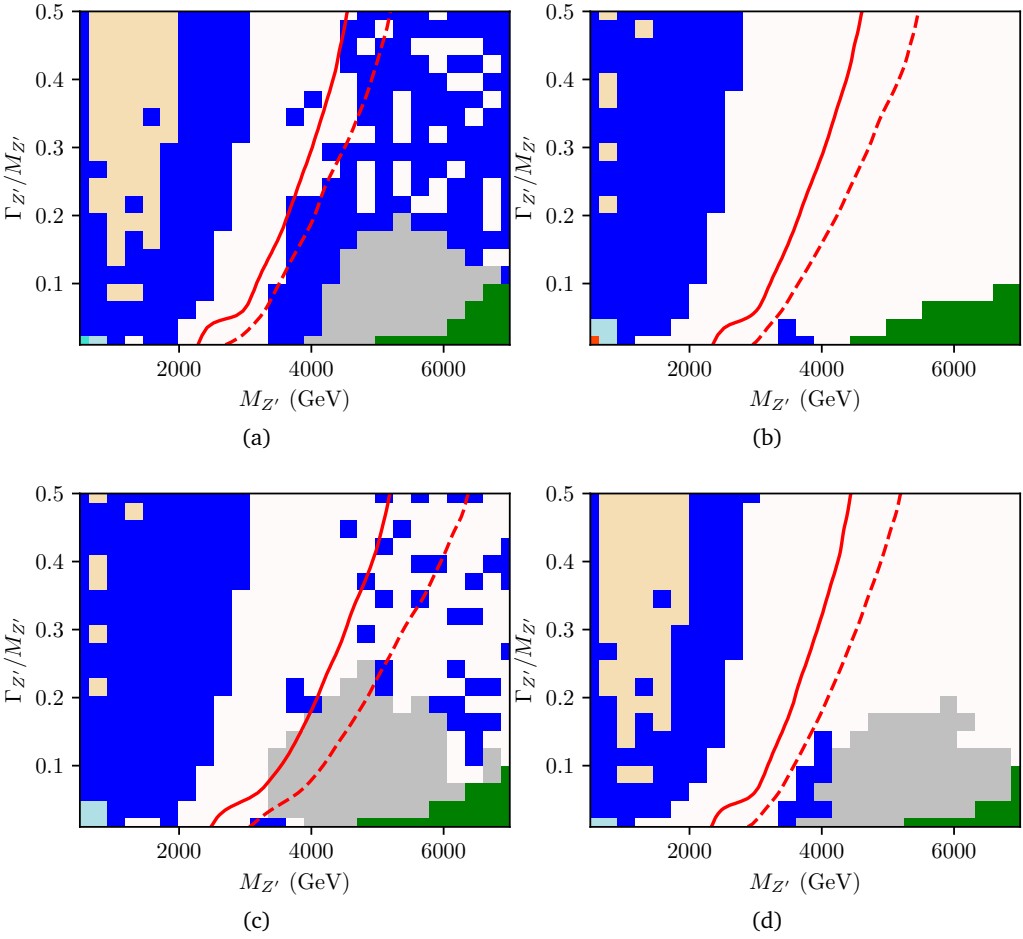

Figure 2: Sensitivity to the leptophobic TC model, in the $Z'$ mass (GeV) versus $\Gamma_{Z'}/M_{Z'}$ plane, for $Z' \to t\bar{t}$. The coloured blocks indicate the most sensitive final state (see legend below). The 95% CL (solid red) and 68% CL exclusion (dashed red) contours are superimposed. (a) Data used as background, but only those measurements with available SM predictions are used. (b) Using NLO SM prediction for background. (c) Using NNLO SM prediction for background. (d) Expected limit using NNLO SM prediction for background.

SM theory agrees reasonably well with the measurement, and the measurement uncertainties dominate given the precision of the SM calculation. The limits in Fig. 2b are somewhat stronger than the default case because in some regions the SM prediction already overshoots the data slightly, so this existing minor discrepancy adds to that caused by injecting an additional BSM contribution, as seen in Fig. 3a and Fig. 3b. In Fig. 2c, NNLO SM $t\bar{t}$ predictions are used for the (13 TeV) SM backgrounds; again the limits are stronger, for example increasing from 4.6 TeV at NLO to 5.2 TeV for NNLO, at 50% $\Gamma_{Z'}/M_{Z'}$. This is due to a reduction in scale uncertainties, as seen in Fig. 3c, and highlights the importance of increased SM precision in extending the reach of the LHC for BSM physics. Finally, in Fig. 2d we show this "expected" limit, evaluated by moving the central value of the measurement to lie exactly on the SM theory prediction, but retaining the measurement uncertainties. We see that the actual limits are slightly stronger than the expectation, again due to the fact that the SM theory lies slightly above the data.

### 3.3 Dijet signature

As discussed above in the Herwig comparison, the LO Herwig calculation is inclusive, and so all decays of the $Z'$ are generated, including those to first generation quarks. This, coupled with the fact that hadronic top decays also lead to jets, leads to the sensitivity at the highest masses being dominated by the ATLAS 13 TeV jet measurements [56], with an improved sensitivity compared to POWHEG, see Fig. 1. This comes principally from contributions to the central dijet invariant mass measurement [56], with the high mass multijet final states [62] playing a minor role. SM predictions for these final states are less precise than for top production, and uncertainties can be at least comparable to those in the data, so the assumption that the SM is identical to the data becomes difficult to justify. For the multijet final states, the state-of-the art predictions are high-multiplicity tree-level calculations matched to parton showers.[2] The spread of such predictions (as shown in [62]) is indeed comparable to the data uncertainties. If the multijet measurements are removed, and only measurements for which more precise predictions are available are used, the sensitivity in Fig. 1b is slightly reduced, to that shown in Fig. 4a.

The dijet measurement, for which an NLO QCD calculation is available [57], is still used in this case. The exclusion due to this measurement, using the data as the background, is illustrated in Fig. 5a for $\cot\theta_{\rm H} = 4.5$ and $M_{Z'} = 3.6$ TeV. However, also shown in that figure is the NLO QCD SM prediction. Not only are the uncertainties comparable to those of the measurement, but the prediction falls below the data at high dijet mass.

The expected exclusion (Fig. 5b) would still be above 95%, but the actual exclusion using the SM prediction as background is zero. The impact of this is that at high $\cot\theta_{\rm H}$ the expected limit, shown in Fig. 4b, is higher than the actual limit shown in Fig. 4c, and the jet cross section measurements are in fact never the most sensitive.

## 4 Discussion and conclusions

The exclusion limits obtained in this analysis are summarised in Tab. 2 where we have used the available measurements in both leptonic and hadronic decay modes, with the maximum integrated luminosity of any measurement being 36.1/fb. As can be seen, we exclude $M_{Z'}$ below 2.29, 3.17 and 4.01 TeV when data are used as background, for widths of 1, 10, and 30% of the mass respectively. For the same width fractions, these numbers become 2.35, 3.22 and 4.04 TeV when the NLO prediction is used for background, and 2.50, 3.55, 4.53 when the

---

[2]Although NNLO calculations for three-jet final states have recently been presented [63], comparisons to these measurements are not yet available.

NNLO predictions are used. Moreover, our scans reaching up to the fraction of $\Gamma_{Z'}/M_{Z'} = 50\%$ exclude it below 4.54 TeV, 4.61 and 5.19 TeV, again for data, NLO and NNLO predictions used for background, respectively.

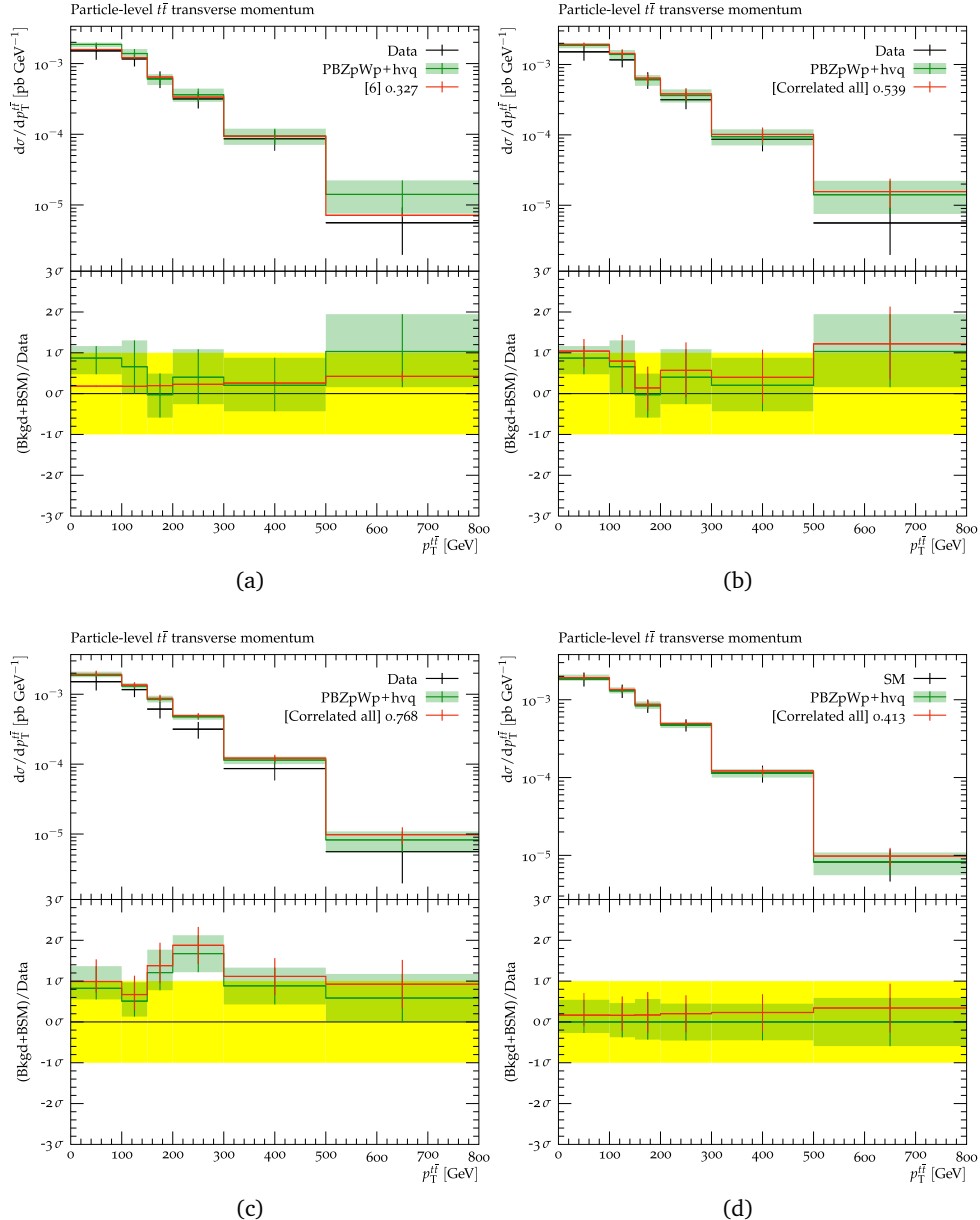

Figure 3: ATLAS all-hadronic boosted $t\bar{t}$ measurement, and `PBZpWp` signal for $M_{Z'} = 4.56$ TeV, $\Gamma_{Z'}/M_{Z'} = 0.5$. Transverse momentum distribution for $t\bar{t}$, (a) using data as background, (b) using NLO SM as background, (c) using NNLO SM as background, (d) Expected exclusion using NNLO SM prediction for background. In each case the black points are the measurement, the red histogram is the SM background + BSM signal, and the green is the SM prediction. The lower insets show the ratio of the signal plus background to the measurement, with the yellow band indicating the combined $1\sigma$ uncertainty on the ratio, and the green band indicating the uncertainty on the SM prediction.

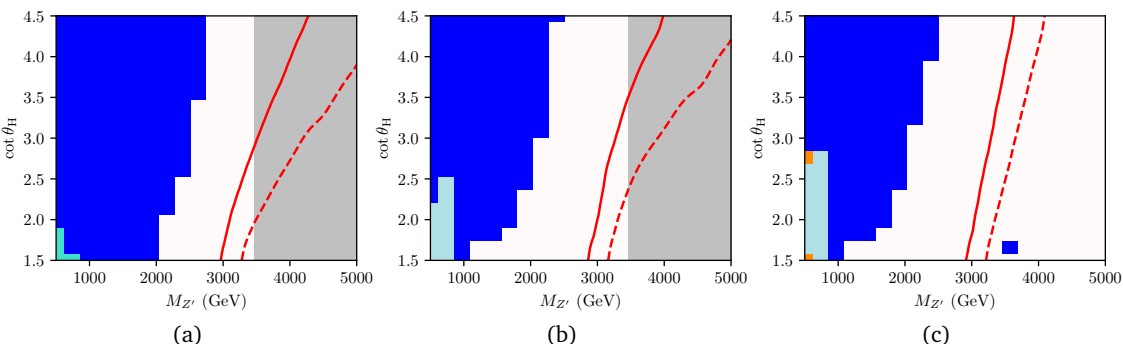

Figure 4: Exclusions derived using Herwig. (a) As Fig. 1b but only using those measurements for which SM predictions are available. (b) Expected limit. (c) Measured limits using the SM predictions as background.

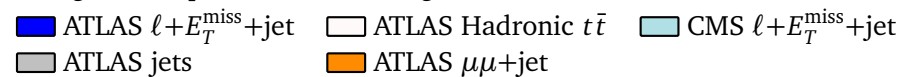

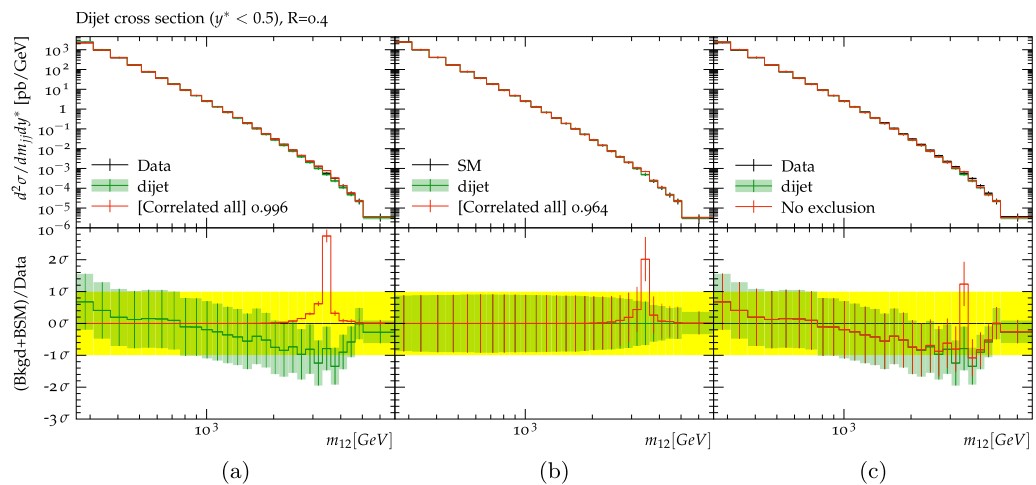

Figure 5: ATLAS jets measurements, and Herwig signal for $M_{Z'} = 3.6$ TeV, $\cot\theta_{\mathrm{H}} = 4.5$. (a) Dijets using data as background. (b) Dijets expected exclusion. (c) Dijets using NLO SM as background. In each case the black points are the measurement, the red histogram is the SM background + BSM signal, and the green is the SM prediction. The lower insets show the ratio of the signal plus background to the measurement, with the yellow band indicating the combined $1\sigma$ uncertainty on the ratio, and the green band indicating the uncertainty on the SM prediction.

The fact that the limits using SM calculations as background are somewhat stronger than those obtained in the default Contur mode when data are used may seem surprising, since the default mode effectively assumes that the SM uncertainties are negligible, whereas the SM uncertainties are correctly accounted for when the calculations are used. It arises, as already mentioned, because the SM prediction lies slightly above the data, so any signal on top of it takes the prediction still further away from the data. The impact of more precise SM predictions is seen in the increased limits when NNLO predictions are used compared to NLO.

Our exclusions can be compared to the strongest limits to date on this model, coming from resonance searches by ATLAS and CMS. CMS [20] excludes the TC $Z'$ boson below 3.80, 5.25,

Table 2: Exclusion limits on $M_{Z'}$ obtained in this analysis.

| | Excluded $M_{Z'}$ [Tev] | | |
|---|---|---|---|
| $\Gamma_{Z'}/M_{Z'}$ [%] | Data as bgd. | NLO as bgd. | NNLO as bgd. |
| 1 | 2.29 | 2.35 | 2.50 |
| 10 | 3.17 | 3.22 | 3.55 |
| 30 | 4.01 | 4.04 | 4.53 |
| 50 | 4.54 | 4.61 | 5.19 |

and 6.65 TeV for 1, 10, and 30% widths respectively, using leptonic and hadronic decays of the top in 35.9/fb of data. ATLAS [17] excludes it below 3.9 and 4.7 TeV for decay widths of 1 and 3% respectively using the fully hadronic decay channel only in 139/fb of integrated luminosity. An earlier ATLAS search [15], using the semileptonic decay mode in 36.1/fb of integrated luminosity excludes the $Z'$ bosons with $M_{Z'}$ below 3 (3.8) TeV for 1% (3%) decay width. In [16], using the fully hadronic decay mode in 36.1/fb of integrated luminosity, ATLAS excludes $Z'$ bosons with mass below 3.1 (3.6) TeV for 1% (3%) decay width.

The limits in our analysis are significantly weaker than the direct searches. Some of this difference comes from the fact that no measurements using the full Run 2 luminosity of the LHC are yet available in Rivet. However, a more significant factor is the binning of the measurements. In a measurement unfolded to particle level, the binning is generally chosen to ensure that there are several events (typically at least of order ten) in each bin. The searches use a binned maximum likelihood fit with no such constraint, and of course the sensitivity at high mass comes from the tails of the distribution, where there are many empty bins.

With Contur we are also able to derive new exclusion limits in a previously unexplored region of the parameter space where $\Gamma_{Z'}/M_{Z'} > 30\%$, a region where direct searches based on bump hunting, without precise SM background calculations, become more challenging.

This analysis therefore illustrates both the strengths and the weaknesses of a Contur-like approach, using differential cross section measurements to constrain BSM physics.

On the one hand, in the regions where the SM cross section is significant, we validate the Contur approach, using either data or SM predictions as background. The advantage of this is that a very wide range of BSM models can be rapidly studied. This advantage becomes very apparent in models with a greater number of free parameters and more complex phenomenology [64–66]. In this sense our results support the assumptions made in such studies.

On the other hand, in this study we have addressed a model with a single, clear signature for which several dedicated searches already exist. In this case, the benefits of a more global analysis are minimal, and the Contur exclusions are not found to be competitive. The greater reach of the searches comes from their use of the low statistics tails of distributions, where particle-level cross section measurements have not yet been made, or have been made with very coarse binning. It is not clear this is a fundamental limitation; upper limits on model-independent cross sections could be used by Contur when provided, and discussions about the best way to publish statistical information from experiments [67,68] should also consider these observables.

Looking to the future, the precision of the measurements, and probably the SM predictions, will increase throughout the high-luminosity LHC period, while no large leaps in energy are anticipated for the foreseeable future. This implies that the relative reach of measurement-based approaches compared to searches seems likely to increase. Meanwhile, the theoretical landscape of BSM ideas continues to grow, increasing the value of making model-independent measurements which can be reinterpreted in multiple scenarios.

## Acknowledgements

We are grateful to L. Corpe and S. Kraml for carefully reading the draft and for useful discussions. We also thank J. Mazzitelli for providing us with the NNLO+PS event samples.

**Funding information**   JMB has received funding from the European Union's Horizon 2020 research and innovation program as part of the Marie Skłodowska-Curie Innovative Training Network MCnetITN3 (grant agreement no. 722104) and from a UKRI Science and Technology Facilities Council (STFC) consolidated grant for experimental particle physics. Work at WWU Münster was funded by the Deutsche Forschungsgemeinschaft (DFG, German Research Foundation) through Project-Id 273811115 - SFB 1225 and the Research Training Network 2149 "Strong and weak interactions - from hadrons to dark matter". The work of TJ was also supported by the DFG under grant 396021762 - TRR 257. The work of MMA and IS was supported in part by the IN2P3 master project Théorie-BSMGA. The Work of MMA is also supported by the National Science Center, Poland, under the research grant 2017/26/E/ST2/00135.

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
