# Peer review of "Probing a leptophobic top-colour model with cross section measurements and precise signal and background predictions: a case study"

_SciPost Physics Core, doi:SciPost Phys. Core 6, 014 (2023)_

## Round 1 · Referee Report · Anonymous (Referee 1) · 2022-2-4

Strengths

Clear demonstration of the impact of using higher-order theory predictions for the Standard Model background in an analysis performed with the CONTUR toolkit.

Weaknesses

Limited scope of the results.

Report

The authors use the CONTUR toolkit to determine exclusion limits for a heavy neutral $Z^\prime$ vector boson from results of the ATLAS and CMS experiments. The $Z^\prime$ is motivated by a particular top-colour model and couples only to first- and third-generation quarks. This leads to a clear $t\bar{t}$ signature and allows the authors to compare their limits to those of existing dedicated searches.

The authors investigate the effects of extending the default CONTUR analysis to include known Standard Model (SM) background predictions at next-to-leading order (NLO) and next-to-next-to-leading order (NNLO) as well as beyond the SM (BSM) signal predictions at NLO. They demonstrate that including NLO and NNLO SM background predictions can considerably change the limits compared to the default CONTUR setup, which assumes that the SM prediction is equal to the experimental data and has negligible uncertainties.

The paper is well written. The analysis is described in detail and the effects of extending the standard CONTUR setup are discussed in a clear and comprehensive manner. The results are interesting, in particular to assess the impact of the simplifying assumption of using the experimental data as SM background. However, apart from this, the determined bounds on the considered top-colour model are of limited scope. In fact, the authors show that existing dedicated searches provide considerably stronger bounds.

I believe that the results of the paper are not groundbreaking enough to meet the strict acceptance criteria of SciPost Physics. Nevertheless, this is a solid paper that I would recommend publishing in SciPost Physics Core. There are some minor points that I list below and would ask the authors to address.

Requested changes

  1. In the abstract, the authors write that they study "fiducial cross section measurements from ATLAS, CMS and LHCb". However, neither Table 1 nor the captions of any of the figures lists an LHCb measurement. I assume "LHCb" should be removed.

  2. In the introduction, the authors mention the LHC schedule and its luminosity targets . The schedule of Run 3 (the authors write "2021-2023") and the HL-LHC (the authors write "starting in 2026"), as well as the luminosity target for Run 3 (the authors write "statistics will be roughly doubled to 300 fb$^{-1}$") seem to be outdated. According to the current LHC schedule, Run 3 will last from 2022-2025, and the luminosity target for Run 3 only is 250 fb$^{-1}$, while the HL-LHC is supposed to start in 2029 (cf. http://lhc-commissioning.web.cern.ch/schedule/LHC-long-term.htm).

  3. At the end of page 2, CONTUR and RIVET are mentioned, but without any explanation of what they actually are and what they are specifically used for (nor what the acronyms mean). I suggest the authors add at least a sentence in the introduction that briefly describes these tools and how they are related to each other.

  4. In Fig. 1 and 4, the y-axis of the plots shows $\cot\theta_H$, while in Fig. 2 it shows $\Gamma_{Z^\prime}/M_{Z^\prime}$. It might be more easy for the reader to compare the plots to each other and also to other searches, if the plots in Fig. 1 and 4 would also use $\Gamma_{Z^\prime}/M_{Z^\prime}$ for the y-axis. Since there is a one-to-one correspondence between $\cot\theta_H$ and $\Gamma_{Z^\prime}/M_{Z^\prime}$ in eq. (2), it seems to be easy to rescale these plots. Is there a particular reason why the authors do not use $\Gamma_{Z^\prime}/M_{Z^\prime}$ in Fig. 1 and 4? If they agree that doing this could improve the clarity, they might want to change the y-axis in Fig. 1 and 4 and use $\Gamma_{Z^\prime}/M_{Z^\prime}$ everywhere.

  5. In Table 1, the authors list the data sets used in their analysis. In Figs. 1, 2, and 4, the different final states contributing to the plots are labelled in the captions. At the moment, there is no clear correspondence between the data sets in Table 1 and the final states in Figs. 1, 2, and 4. In addition, according to the discussion in section 3.3 (cf. also Fig. 4), precise SM predictions are not available for all of the measurements. This is not clear from Table 1. At the moment, there is only the column "Highest SM Order", which suggest that SM predictions of at least NLO are available for all data sets. I recommend that the authors extend Table 1 as follows:

  6. It would be useful to see which final states (i.e. those shown by different colours in Figs. 1,2,4) correspond to which data set in Table 1.
  7. It would be useful to see the precision of the SM predictions for the various measurements contained in the data sets in Table 1 (e.g. NLO SM for dijets but less precise measurements for multijets).

  8. At the bottom of page 6, the authors write that the Herwig analysis shows improved sensitivity compared to POWHEG "at low $\cot \theta_H$" and they refer to Fig. 1. However, in Fig.1 it seems like the sensitivity is improves more at large $\cot \theta_H$ than at low $\cot \theta_H$. They authors might want to clarify their statement.

  9. Some references of measurements mentioned in the text are not listed in Table 1, e.g. Refs. [57] and [58]. This might be related to point 5 above. However, in addition, Ref. [58] seems to be identical to Ref. [55].

  10. At the end of the first paragraph on page 11, the authors write that in Fig. 4a, the 95% exclusion stops "just below 4 TeV, rather than just above it". However, for the largest value of $\cot \theta_H$ shown in Fig. 4a, the corresponding line actually reaches above 4 TeV.

  11. In the next-to-last paragraph on page 13, it seems like there is an "of" missing in "greater number [of] free parameters".

  • validity: high
  • significance: good
  • originality: good
  • clarity: high
  • formatting: excellent
  • grammar: perfect

Author:  Mohammad Mahdi Altakach  on 2022-04-21  [id 2399]

(in reply to Report 1 on 2022-02-04)

We would like to thank the referee for his report and apologise for the delay.

Please find below our answers to the requested changes (the changes (red coloured) can be checked in the attached version of the paper):

1- We added the LHCb measurement that we consider in our analysis to Table 1. The reason why it is not listed in any of the figures is that it was never the most sensitive one (this is explained now as a foot note in (red)).

2- We have corrected these information (red colour in the text).

3- We've added a couple of sentences (red colour in paragraph number 4) to briefly explain Contur and Rivet.

4- Fig. 2 uses only data and SM predictions, which are independent of \cot\theta_H. We therefore prefer to show this plot in the physical parameter plane m_ZP and Gamma_ZP/m_ZP.

5- We've added the labels (pools) of the measurements of the plots where we have simulated predictions to Table 1 (red). So now in the first column of table 1, in each box we have the measurement on the first line and its corresponding pool on the second line. Each pool contains several measurements, and in table 1 we only include the measurements that we have theory prediction for. Concerning the precision of the SM predictions, all the measurements that we have theory predictions for are listed in Table 1 with their references and their precisions, if a measurement is not listed in Table 1, it means that we don't have a theory prediction for it. This is why we include ATLAS 2017 I1609448 measurement (ref [56]) and we don't show ATLAS multijet measurement [62].

6- We've adjusted paragraph 1 of section 3.3 accordingly (in red).

7- We fixed the issue of the identical references. Concerning the first part of comment 7, As we clarified in 5, all the measurements that we have theory predictions for are listed in Table 1 with their references, if a measurement is not listed in Table 1, it means that we don't have a theory prediction for it.

8- Fixed, please see the text in red at the end of paragraph one of section 3.3.

9- Fixed

Attachment:

Contur_TC_paper.pdf

Anonymous on 2022-05-05  [id 2445]

(in reply to Mohammad Mahdi Altakach on 2022-04-21 [id 2399])

The authors have addressed all the requested changes in their comment. The only remaining remarks that I have are related to the following points:

(4) My suggestion was actually not to change the plot in Fig. 2 but to use the physical parameter plane m_ZP and Gamma_ZP/m_ZP not only in Fig. 2 but also in Figs. 1 and 4. This would make it easier to compare the plots to each other and to those of other Z' searches.

(5)/(7) The authors made it more clear that only measurements with at least NLO theory predictions are included in table 1. They now write in the last sentence of section 2.2:
"For convenience, all the employed data sets that we have theory predictions for are summarised in Tab. 1."
However, as described by the authors in section 3.3, theory predictions are also available for multijet final states in terms of "high-multiplicity tree-level calculations matched to parton showers". To avoid confusion about why the corresponding measurements are not included in table 1, further clarification might be useful, e.g. by mentioning "higher-order theory predictions" or "NLO and NNLO theory predictions" in the last sentence of section 2.2. A similar comment could be useful in the caption of table 1.

Anonymous on 2022-05-23  [id 2505]

(in reply to Anonymous Comment on 2022-05-05 [id 2445])

Concerning question 4, we understand the suggestion of the referee, but for technical reasons it isn't doable unfortunately.

(5)/(7): As suggested by the referee, we changed the last sentence in section 2.2 to:
"For convenience, all the employed data sets that we have higher-order theory predictions for are summarised in Tab. 1. "
And the caption of Tab. 1 to:
" Table of the Rivet routines used for the limit-setting scan. Only routines where we have higher-order theory predictions are listed. The number in the Contur category indicates the centre-of-mass energy. The calculations are performed by the authors unless otherwise cited."

---

## Round 2 · Referee Report · Anonymous (Referee 1) · 2022-10-21

Strengths

Clear demonstration of the impact of using higher-order theory predictions for the Standard Model background in an analysis performed with the CONTUR toolkit.

Weaknesses

Limited scope of the results.

Report

This paper was initially submitted to SciPost Physics. The previous referee report recommended it for publication in SciPost Physics Core and asked the authors to address a number of minor points (see https://scipost.org/submissions/2111.15406v1/#report_1).

The authors have implemented all the requested changes and have commented on them in their answers to the initial referee report. All points raised in the initial report have been addressed in a satisfactory manner. This has further improved the paper and I recommend it for publication in SciPost Physics Core.

---

## Round 2 · Referee Report · Anonymous (Referee 2) · 2022-10-26

Strengths

Detailed study of the impact of NNLO SM predictions to extract exclusion limits on BSM physics

Weaknesses

  1. General analysis limited to a single specific model
  2. Exclusion limits significantly weaker than those of dedicated direct searches

Report

The authors test the CONTUR kit ability in putting exclusion limits on the parameter space of an extra neutral Z' (mass, width), using data collected by the ATLAS and CMS experiments at LHC. Due to the selective leptophobic character of the new gauge vector boson, the signal arises from Z' hadronic decays.

The authors implement different estimates of the SM background within the CONTUR framework. The standard approach of CONTUR (SM predictions matching the existing data), is compared to theory-driven estimates (SM background evaluated from the theory at NLO or at NNLO). The impact of more precise SM predictions is seen in the increased limits when NNLO predictions are used compared to NLO and to the standard CONTUR setting.

The paper is well written. The analysis is clearly and comprehensively described. The results are interesting, but weakened by the significantly better limits obtained by the experimental collaborations in dedicated direct searches. In my opinion the results of the paper are not sufficiently innovative to meet the strict acceptance criteria of SciPost Physics. Nevertheless, the paper contains interesting enough information to allow publication in SciPost Physics Core.

Requested changes

The paper can be published in its present form.

---

## Editorial Decision

published